# PET Imaging of Neuro-Inflammation with Tracers Targeting the Translocator Protein (TSPO), a Systematic Review: From Bench to Bedside

**DOI:** 10.3390/diagnostics13061029

**Published:** 2023-03-08

**Authors:** Ferdinando Corica, Maria Silvia De Feo, Joana Gorica, Marko Magdi Abdou Sidrak, Miriam Conte, Luca Filippi, Orazio Schillaci, Giuseppe De Vincentis, Viviana Frantellizzi

**Affiliations:** 1Department of Radiological Sciences, Oncology and Anatomo-Pathology, Sapienza University of Rome, 00185 Rome, Italy; 2Department of Nuclear Medicine, Santa Maria Goretti Hospital, 04100 Latina, Italy; 3Department of Biomedicine and Prevention, University Tor Vergata, 00133 Rome, Italy

**Keywords:** TSPO, translocator protein, PET, neuro-inflammation, peripheral benzodiazepine receptor

## Abstract

Parkinson’s disease is the second most common neurodegenerative disorder, affecting 2–3% of the population of patients >65 years. Although the standard diagnosis of PD is clinical, neuroimaging plays a key role in the evaluation of patients who present symptoms related to neurodegenerative disorders. MRI, DAT-SPECT, and PET with [^18^F]-FDG are routinely used in the diagnosis and focus on the investigation of morphological changes, nigrostriatal degeneration or shifts in glucose metabolism in patients with parkinsonian syndromes. The aim of this study is to review the current PET radiotracers targeting TSPO, a transmembrane protein that is overexpressed by microglia in another pathophysiological process associated with neurodegenerative disorders known as neuroinflammation. To the best of our knowledge, neuroinflammation is present not only in PD but in many other neurodegenerative disorders, including AD, DLB, and MSA, as well as atypical parkinsonian syndromes. Therefore, in this study, specific patterns of microglial activation in PD and the differences in distribution volumes of these radiotracers in patients with PD as compared to other neurodegenerative disorders are reviewed.

## 1. Introduction

The 18kDA translocator protein (TSPO), previously known as peripheral benzodiazepine receptor (PBR), is a transmembrane protein located on the outer mitochondrial membrane (OMM), and it is mainly expressed by glial cells (e.g., Astrocytes, microglia) and endothelial cells in the CNS. Its expression is strongly upregulated in activated microglial cells by inflammatory stimuli, and it’s implicated in the progression of many neuropsychiatric disorders such as Alzheimer’s Disease, Parkinson’s Disease, amyotrophic lateral sclerosis, and multiple sclerosis as a part of a process known as neuroinflammation [1,2,3,4,5]. Neuroinflammation is defined as an inflammatory response mediated by the release of pro-inflammatory cytokines, chemokines, ROS, secondary messengers, and overexpression of proteins, such as TSPO, in the brain or in the spinal cord [6]. The overexpression of TSPO in several neurodegenerative disorders makes it a possible biomarker for neuroinflammation, and in this setting, the in vivo quantification of TSPO expression in microglial cells using PET radioligands could offer important insights into the pathogenesis and onset of diseases such as PD and, consequently, on early treatment of such diseases [3,7,8]. To date, several radioligands targeting TSPO have been developed and studied in clinical and pre-clinical studies in patients with neurodegenerative disorders. (R)-^11^C-PK11195 was the first developed radioligand targeting TSPO, showing disease-related TSPO expression in several neurodegenerative diseases. However, the poor signal-to-noise ratio of this radioligand has led to the development of 2nd generation radioligands, such as [^11^C]-PBR28, [^18^F]-FEPPA, [^18^F]-DPA714, and [^11^C]-DPA713 [9]. Second-generation radioligands targeting TSPO have a better signal-to-noise ratio than (R)-[^11^C]-PK11195, but they show a large inter-subject variability due to a single nucleotide polymorphism (SNP) (rs6971) in the gene encoding TSPO. Because of the presence of this polymorphism, three classes of binding affinity are defined: homozygote high-affinity binders (HABs), homozygote low-affinity binders (LABs), heterozygote mixed affinity binders (MABs) [10]. To date, third-generation radioligands have been developed in recent years, such as [^11^C]ER-176 and [^18^F]-GE-180, which show improved binding to TSPO in LABs. The aim of this systematic review is to provide an overview of the existing literature regarding PET radioligands targeting TSPO in patients affected by Parkinson’s Disease in order to understand which are the goals and future perspectives in terms of early diagnosis and treatment of neuroinflammation and its implication in the pathogenesis of PD by means of correlation between neuroinflammation and other pathogenic pathways as the dopaminergic pathway damage and β-amyloid deposition. 

## 2. Materials and Methods

### 2.1. Search Strategy and Study Selection

Research until November 2022 was performed in Pubmed, Scopus, Web of Science, and Cochrane Central databases to retrieve clinical studies concerning the use of PET radioligands targeting TSPO in patients affected by PD. The research was carried out according to PRISMA guidelines using the terms: (1) TSPO or “translocator protein”; (2) PET or positron emission tomography; (3) Parkinson’s or Parkinson’s disease or PD; (4) neuroinflammation. Only clinical studies in the English language were included. Case reports, short communications, and editorials were excluded (See Figure 1). 

### 2.2. Data Extraction and Methodological Quality Assessment

Authors’ generalities, country and year of publication, and demographic and clinical characteristics of patient populations were collected. A clinical appraisal skill program (CASP) was used to evaluate the diagnostic accuracy of the clinical studies evaluated in this systematic review.

## 3. Results

### 3.1. Analysis of the Evidence

The resulting PRISMA search strategy is shown in Figure 1. Thirteen articles were selected from the systematic review of the literature, with an overall number of 396 patients, including 223 patients affected by PD, 83 patients affected by other neurodegenerative diseases (66 affected by multiple system atrophy, 11 by atypical parkinsonian syndromes and six by dementia with Levy bodies), and 90 healthy subjects. The main characteristics of the manuscripts taken into account are listed in Table 1. The selected studies’ quality appraisal is shown in Table 2.

#### 3.1.1. (R)-^11^C-PK11195 

A retrospective study conducted by Ouchi et al. comparing the distribution of (R)-^11^C-PK11195 with that of [^11^C]-CFT, a dopamine transporter ligand used to determine dopamine hypofunction in normal aging, PD, AD, and other neurological disorders [24], showed higher accumulation of the first in the midbrain of patients with early, drug-naive PD patients with respect to healthy subjects, suggesting that microglial activation and overexpression of pro-inflammatory cytokines by microglial cells causing neuroinflammation is more significant in patients with PD with severe nigrostriatal pathway damage. This observation was supported by the increased uptake of (R)-^11^C-PK11195 in the midbrain as compared with the decreased accumulation of [^11^C]CFT in the dorsal putamen [11]. Kobylecky et al. conducted a study with patients affected either by idiopathic PD or atypical parkinsonian syndromes (multisystemic atrophy or progressive supranuclear palsy). In this study, the patients underwent both diffusion-weighted MR imaging and PET scan with an injected dose of 296 MBq (R)-^11^C-PK11195. As a result, it was observed an increased binding potential of (R)-^11^C-PK11195 in the caudate, putamen, midbrain and pons of patients with atypical parkinsonian syndromes and in the pons of patients with PD. Nevertheless, no correlation was found between (R)-^11^C-PK11195 binding potential and disease severity and duration in both groups [12]. Edison et al. studied the relationship between microglial activation, amyloid and glucose metabolism in patients with PD dementia (PDD) and PD without dementia. Such patients underwent PET with (R)-^11^C-PK11195, [^11^C]PiB and [^18^F]-FDG. The analysis of (R)-^11^C-PK11195 PET data showed a statistically significant (35–65%) increase in the binding potential (BP) of the radioligand, in patients with PDD, in the anterior cingulate gyrus, posterior cingulate gyrus, thalamus, striatum, frontal, parietal and temporal cortex, while in patients with PD, the increase in BP was statistically significant in the temporal and frontal cortices. Also, an inverse correlation was found between Mini-Mental State Examination (MMSE) scores and cortical microglial activation [13]. Iannaccone et al. compared the binding potential of (R)-^11^C-PK11195 in patients with early dementia with Levy bodies (DLB) and PD., observing an increased BP in the substantia nigra and the putamen of both groups. Furthermore, few PD patients also showed an increased BP in the anterior cingulate and medial prefrontal regions [14]. The same radiotracer was used as an in vivo marker of peripheral benzodiazepine site expression in patients with idiopathic PD in a longitudinal study by Gerhard et al. In this study, a significant increase in the uptake of (R)-^11^C-PK11195 was observed in the subcortical mean, including the striatum, pallidum, thalamus and pons, and in cortical regions including the precentral gyrus, frontal lobe, anterior and posterior cingulate gyrus [15]. Interestingly, it was observed how, in 8 patients enrolled in this study who underwent PET with (R)-^11^C-PK11195 after 18–24 months, the distribution and concentration of (R)-^11^C-PK11195, hence the microglial activation, was not different from baseline, suggesting that microglial activation occurs early in the neuroinflammation process and is then maintained by cytokine release over time.

#### 3.1.2. [^18^F]-FEPPA

In 2015 Koshimori et al. first studied the utility of [^18^F]-FEPPA as a biomarker for neuroinflammation in patients with PD. In addition, this study aimed to determine the effects of the rs6971 SNP on the BP of [^18^F]-FEPPA in patients with PD and the correlation between TSPO binding and clinical measures of PD. As a result, it was seen that PD-HAB showed an increase in TSPO binding in the caudate nucleus and putamen as compared to PD-MAB patients, although the disease effect on TSPO binding was not statistically significant in such regions. Furthermore, it has been seen that the V_t_ of [^18^F]-FEPPA doesn’t correlate with the use of anti-parkinsonian medication, disease duration and Unified Parkinson’s Disease Rating Scale (UPDRS) score [16]. Similar results were observed in a subsequent study using [^18^F]-FEPPA by Ghadery et al. Indeed, also, in this case, it was observed that PD-HAB patients had higher V_t_ values in all brain regions compared to PD-MABs. Similarly, no correlation was found between the V_t_ values, levodopa equivalent doses (LEDD) and UPDRS scores, as well as disease duration, in both PD-HABs and PD-MABs. Since this study enrolled patients with normal cognitive impairment, it wasn’t possible to demonstrate an anatomically widespread microglial activation or neuroinflammation in primary cortical regions, suggesting that such patterns may be seen in patients with a more severe cognitive impairment [17]. Finally, they studied in a dual-tracer PET study using [^11^C]PiB, a thioflavin-T analog that at radiotracer concentrations has a high affinity for fibrillar Aβ [25], and [^18^F]-FEPPA the relationship between brain β-amyloid and neuroinflammation in patients affected by PD with mild (PD-MCI) or without (PDn) cognitive impairment, observing that, in patients within the PD-MCI group who were PIB-positive, the ^18^F-FEPPA V_t_ was increased in the frontal and temporal lobes, striatum, precuneus and dorsolateral prefrontal cortex. This pattern was also observed in the frontal lobe of PDn patients. A correlation between disease duration, PIB positivity, and FEPPA V_t_ was also found. Indeed, patients with short disease duration and PIB-positive showed a significantly high [^18^F]-FEPPA V_t_ in the precuneus, temporal and frontal lobe. A significant association between [^11^C]-PIB distribution volume ratio (DVR) and ^18^F-FEPPA V_t_ was observed in the ventrolateral prefrontal cortex, medial prefrontal cortex, dorsolateral prefrontal cortex and striatum, suggesting that a higher β-amyloid deposition correlates with increased microglial activation [18].

#### 3.1.3. ^11^C-PBR28

Jucaite et al. estimated the effects of treatment with the myeloperoxidase inhibitor AZD3241, comparing the V_t_ of [^11^C]-PBR28 at baseline and after 8 weeks of treatment in patients with PD while investigating neurodegeneration in the dopaminergic system undergoing PET with [^18^F]-FE- PE21. In this study, [^11^C]-PBR28 V_t_ values were similar at baseline in all brain regions, with a slight increase in the thalamus. As seen in the previous studies discussed in this systematic review, the individual V_t_ values in this study were dependent on the TSPO genotype. The administration of AZD3241 resulted in a statistically significant reduction of V_t_ at 4 weeks (15.5–16.2%) and at 8 weeks (13.2–15.7%) in the nigrostriatal pathway as compared to baseline, supporting the hypothesis that the inhibition of myeloperoxidase (MPO) has significant effects on neuroinflammation [19]. In 2018 Varnas et al. studied the relationship between dopaminergic pathology and neuroinflammation, and therefore TSPO binding, performing PET studies of DAT using [^18^F]-FE-PE21 and of TSPO binding using [^11^C]-PBR 28. Similarly to previously discussed studies, the effect of the rs6971 SNP was observed since HABs showed an increased V_t_ compared to MABs and LABs. Conversely, the authors didn’t find any correlation between the binding parameters of [^18^F]-FE-PE21, which were markedly decreased in the nigrostriatal pathway, and [^11^C]-PBR28, suggesting that dopaminergic pathology and TSPO binding are independent factors in patients with PD [20]. Recently, Jucaite et al. compared the amount of [^11^C]-PBR28 TSPO binding in patients with multiple system atrophy (MSA) and PD. In this study, the authors were able to demonstrate the presence of a conspicuous pattern of elevated [^11^C]- PBR28 TSPO binding with hotspots in the lentiform nucleus and cerebellar white matter in patients with MSA, whereas such features were not observed in patients with PD. This allowed discrimination between the two pathologies, which often present with overlapping features, with a specificity of 100% and a sensitivity of 83% [21]. 

#### 3.1.4. [^11^C]-DPA-713

Terada et al. studied the progression of microglial activation in patients with early PD (Hoehn and Yahr stage 1 or 2) undergoing PET after administration of 5 MBq/kg of [^11^C]-DPA-713. At the first PET examination, PD patients showed an increased expression of [^11^C]-DPA-713 in the occipital, temporal and parietal cortex, including the brain stem and the basal ganglia, as compared with the control group, with clusters including the left fusiform and precentral gyrus. At the second examination, performed 1 year after the first, PD patients showed an increase in [^11^C]-DPA-713 uptake predominantly in the occipital and parietal cortex, with clusters in left middle frontal, left precuneus, left inferior temporal, left parahippocampal, right inferior occipital, right postcentral, and right superior parietal gyrus. No asymmetry in the BP of [^11^C]-DPA-713 was found [22]. 

#### 3.1.5. [^18^F]-DPA-714

In a study aimed at the designation of an image-derived input function method for [^18^F]-DPA-714, Fang et al. observed that in PD patients, the TSPO genotype critically affects the V_t_ of this radioligand, with HAB subjects, showing increased uptake in [^18^F]-DPA-714 in terms of SUV in respect to MAB subject in the putamen, caudate, thalamus, hippocampus, frontal, temporal, occipital and parietal lobes. Also, the authors stated that since the specific binding of this radioligand in the cerebellum is not negligible, the latter cannot be used as a reference region for TSPO quantification as it may hide the effects of TSPO overexpression in certain pathological conditions, including PD [23]. 

## 4. Discussion

It has been widely demonstrated how neuroinflammation is strongly associated with several neurological and neurodegenerative disorders. TSPO, a transmembrane protein expressed on the OMM, is overexpressed by microglial cells in such disorders and could therefore be targeted as a biomarker for brain injury and neuroinflammation. In this context, we reviewed the existing literature focusing on the development and utility of radiopharmaceuticals binding TSPO in patients affected by PD and how the quantification of microglial activation through PET imaging can affect the diagnosis, treatment and follow-up of patients affected by these neurodegenerative disorders. Moreover, the correlation between TSPO expression and other pathogenic pathways in PD has been examined, as well as the difference in TSPO expression and other neurodegenerative disorders. Important consideration must be done in the analysis of the selected papers: with the exception of the 1st generation radioligand (R)-^11^C-PK11195, the affinity with which the 2nd generation radioligands bind to the TSPO protein is strongly affected by the rs6971 SNP since all the studies examined showed that the uptake in both PD patients and healthy subjects is significantly higher in HABs in respect to MABs and LABs. 3rd generation radiopharmaceuticals targeting TSPO, [^11^C]ER-176 and [^18^F]-GE-180, may improve the detection of TSPO regardless the rs6971 SNP as seen in the first small human studies. Indeed, according to Zanotti-Fregonara et al., [^11^C]ER-176, which is a quinazoline analog of PK11195, showed that its BP for LABs was only one-third lower than HABs [26]. Moreover, Fujita et al. observed that the BP of [^11^C]ER-176 is comparable to that of PBR28 in HABs. In addition, [^18^F]GE-180 showed in a small clinical study involving patients with MS no difference in binding between HABs, MABs and LABs [27]. Although the usefulness of these radiotracers in PD has not been investigated yet, while only a few small clinical studies in patients with other neurological disorders have been carried out, they could represent an improvement in TSPO imaging according to their decreased sensitivity to the rs6971 SNP. In all of the studies taken into account in this systematic review, it has been observed how TSPO expression is increased in patients with PD in any stage as compared to healthy subjects, supporting the observation that neuroinflammation and, therefore, microglial activation play an important role in the pathogenesis of PD. On the other hand, not all the studies taken into account finally demonstrated a significant correlation between TSPO expression and disease duration and severity. This is true for most of the radioligands studied in this systematic review except for [^11^C]-DPA713. In fact, Terada et al. studied the distribution volume of this radioligand at baseline and, after 1 year, observed an increased and more diffuse [^11^C]-DPA713 expression at the second examination in the occipital and parietal cortex [22]. The same outcome was not reached by Gerhard et al., who studied eight patients with PD who underwent PET with (R)-^11^C-PK11195 at baseline and 18–24 months later, without observing significant differences in microglial activation, suggesting that neuroinflammation is a process that occurs in an early phase of PD’s pathogenesis and is then self-maintained over time [22]. Nevertheless, in patients with mild cognitive impairment, as stated in the study by Koshimori et al., it wasn’t observed an anatomically widespread distribution of TSPO or in the primary cortex regions [16]. Future studies with larger cohorts of patients with PD at different stages can be helpful in overcoming such discrepancies and ascertaining if the amount of TSPO expression is correlated to disease severity and duration. In this setting, we observed how dual-tracer PET studies can better correlate neuroinflammation and other physiopathological processes in PD, such as nigrostriatal pathway damage and ß-amyloid deposition. It is still uncertain whether neuroinflammation is correlated to nigrostriatal damage or is an independent factor in the pathogenesis of PD since some discrepancies are seen in different studies involving radiopharmaceuticals targeting TSPO and the nigrostriatal pathway. Indeed, Ouchi et al. observed an inverse correlation between [^11^C](R)-PK11195 and [^11^C]-CFT uptake since patients with more severe nigrostriatal pathway damage, and therefore with a decreased uptake of [^11^C]-CFT, were also showing increased uptake of (R)-^11^C-PK11195 [11]. Conversely, the same correlation was not observed when dual-tracer imaging was performed with ^18^F-FE-PE21 and ^11^C-PBR 28. [19] However, also in the latter, the TSPO imaging with ^11^C-PBR 28 is dependent on the rs6971 SNP, making it more difficult to ascertain a statistically significant correlation between the two pathogenic pathways considering both HABs and LABs patients, while the use of (R)-[^11^C]-PK11195, which is independent of rs6971 polymorphism, and [^11^C]-CFT showed that more severe nigrostriatal damage is associated with an increased TSPO expression. Similarly, it has been observed a significant association between the distribution of volumes of [^11^C]-PIB and [^18^F]-FEPPA in the most recent study by Ghadery et al. [18] The correlation between TSPO overexpression by microglial cells and other pathological features of PD and other neurodegenerative disorders can, eventually, provide an additional tool in the future for the use of radiotracers targeting TSPO, as well as in the study of PD pathogenesis and, eventually, in the therapeutic approach to PD patients. Interestingly, some authors also compared the distribution patterns of TSPO radioligands in patients with PD and other neurodegenerative disorders; the observation that increased uptake of ^11^C-PBR 28 is present in the lentiform nucleus and cerebellar white matter in patients with MSA, while it is not appreciable in patients with PD, can be important for differential diagnosis [21]. Similar outcomes were seen when the uptake of (R)-[^11^C]-PK11195 was compared between patients with PD and DLB; in this case, both groups of patients showed increased uptake in the substantia nigra and putamen, but few PD patients also showed increased uptake in the anterior cingulate and medial prefrontal regions [14]. Finally, the response to therapy can also be a future perspective of radioligands targeting TSPO. While in the studies regarding [^18^F]-FEPPA, no correlation was found between the distribution volume of this radioligand and anti-parkinsonian medications nor with clinical scores, it has been seen that after 4 and 8 weeks of treatment with MPO inhibitors, the distribution volume of ^11^C-PBR28 was significantly decreased as compared to placebo, suggesting that the response to therapies targeting neuroinflammation can be assessed using this radioligand [19]. 

## 5. Conclusions

In conclusion, PET imaging with radioligands targeting TSPO can offer an additional tool in the future for what concerns the management of patients with PD. While other radiopharmaceuticals are already used in the diagnosis of PD, in the future, PET radiotracers for TSPO may be used not only as diagnostic tools but also for prognostic and research purposes. Indeed, TSPO imaging could provide further information on the role of neuroinflammation in the pathogenesis of PD and neurodegenerative diseases as a whole. Indeed, while it is clear that neuroinflammation plays a role in neurodegenerative disorders, it is not yet ascertained how this inflammatory process correlates with the main features of such diseases in terms of timing and severity. Consequently, PET-TC using radioligands targeting TSPO can also serve as an instrument to assess response to therapy and, eventually, be used in the research of new therapies. Moreover, it has been observed how TSPO expression can differ between PD and different neurological disorders in terms of distribution volume and patterns. In this context, TSPO imaging can be useful in the differential diagnosis of neurological disorders according to the amount of TSPO expression and its distribution pattern in the brain. However, some limitations are given by the rs6971 SNP since most of the radiotracers developed show significant results only in HABs patients, considering the low or absent distribution volume in LABs patients, and studies with third-generation radiotracers are still non available in patients with PD. Finally, with additional studies with larger samples and the development of third-generation radiotracers that can overcome the rs6971 polymorphism, PET radiotracers targeting TSPO have a promising future in the study of PD and other neurodegenerative disorders.

## Figures and Tables

**Figure 1 diagnostics-13-01029-f001:**
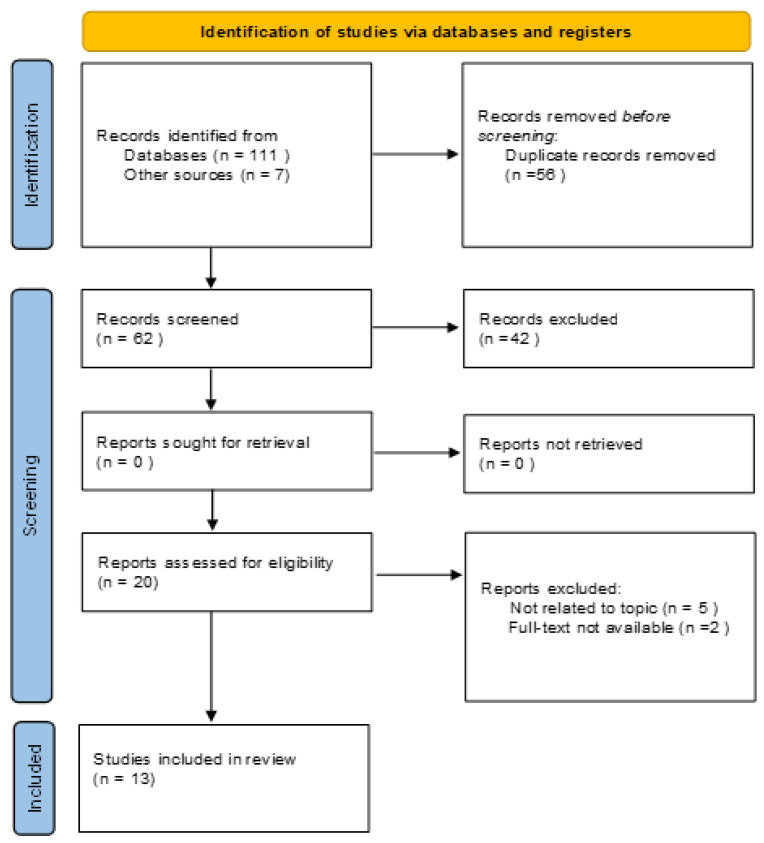
PRISMA flow-chart.

**Table 1 diagnostics-13-01029-t001:** Studies characteristics.

Author	Year of Publication	Country	Tracer	Population	Characteristics
Ouchi et al. [11]	2005	Japan	^11^ C-(R)PK11195	20 patients	10 Parkinson’s Disease 10 Healthy subjects
Kobylecky et al. [12]	2013	UK	^11^ C-(R)PK11195	20 patients	11 Atypical Parkinsonian syndromes 9 Parkinson’s Disease
Edison et al. [13]	2012	UK	^11^ C-(R)PK11195; ^11^ C-PIB; ^18^ F-FDG	19 patients	19 Parkinson’s Disease
Iannaccone et al. [14]	2012	Italy	^11^ C-(R)PK11195	12 patients	6 Parkinson’s Disease 6 Dementia with Levy Bodies
Gerhard et al. [15]	2006	UK	^11^ C-(R)PK11195	29 patients	18 Parkinson’s Disease 11 Healthy subjects
Koshimori et al. [16]	2015	Canada	^18^ F-FEPPA	36 patients	19 Parkinson’s Disease 17 Healthy Subjects
Ghadery et al. [17]	2017	Canada	^18^ F-FEPPA	52 patients	30 Parkinson’s Disease 22 Healthy subjects
Ghadery et al. [18]	2020	Canada	^18^ F-FEPPA; ^11^ C-PIB	41 patients	29 Parkinson’s Disease 12 Healthy subjects
Jucaite et al. [19]	2015	Sweden; Finland	^11^ C-PBR28; ^18^ F-FE-PE21	29 patients	29 Parkinson’s Disease
Varnas et al. [20]	2019	Sweden	^11^ C-PBR28	32 patients	16 Parkinson’s Disease 16 Healthy subjects
Jucaite et al. [21]	2022	Sweden	^11^ C-PBR28	90 patients	66 Multiple System Atrophy 24 Parkinson’s Disease
Tarada et al. [22]	2016	Japan	^11^ C-DPA713	11 patients	11 Parkinson’s Disease
Fang et al. [23]	2022	USA	^18^ F-DPA714	5 patients	3 Parkinson’s Disease 2 Healthy subjects

**Table 2 diagnostics-13-01029-t002:** Quality appraisal.

	1. Was There a Clear Question for the Study to Address?	2. Was There a Comparison with an Appropriate Reference Standard?	3. Did All Patients Get the Diagnostic Test and Reference Standard?	4. Could the Results of the Test Have Been Influenced by the Results of the Reference Standard?	5. Is the Disease Status of the Tested Population Clearly Described?	6. Were the Methods for Performing the Test Described in Sufficient Detail?	7. What Are the Results?	8. How Sure Are We about the Results? Consequences and Cost of Alternatives Performed?	9. Can the Results Be Applied to Your Patients/the Population of Interest?	10. Can the Test Be Applied to Your Patient or Population of Interest?	11. Were All Outcomes Important to the Individual or Population Considered?	12. What Would Be the Impact of Using This Test on Your Patients/Population?
Ouchi et al. 2005 [11]	☺	☺	☺	?	☺	☺	☺	?	☺	☺	☺	It may show patterns of neuroinflammation in patients with PD.
Kobilecky et al. 2013 [12]	☺	☺	☺	?	☺	☺	☺	☺	☺	☺	☺	It may correlate diffusion-weighted images and PET images in patients with PD.
Edison et al. 2012 [13]	☺	☺	☺	☺	☺	☺	☺	☺	☺	☺	☺	Comparison of different radiotracers in PD.
Iannaccone et al. 2012 [14]	☺	☺	☺	☺	☺	☺	☺	?	☺	☺	☺	Comparison of neuroinflammation patterns in PD and DLB.
Gerhard et al. 2006 [15]	☺	☺	☺	☺	☺	☺	☺	☺	☺	☺	☺	It may explain the self-maintaining process of neuroinflammation over time.
Koshimori et al. 2015 [16]	☺	☺	☺	☺	☺	☺	☺	?	?	?	☹	Differentiates binding potential between HABs and MABs.
Ghadery et al. 2017 [17]	☺	☺	☺	☺	☺	☺	☺	?	?	?	☺	Differentiates binding potential between HABs and MABs and correlate the binding potential with the severity of the disease.
Ghadery et al. 2020 [18]	☺	☺	☺	☺	☺	☺	☺	☺	?	?	☺	Studies the relationship between brain β-amyloid and neuroinflammation in patients affected by PD according to their disease severity.
Jucaite et al. 2015 [19]	☺	☺	☺	☺	☺	☺	☺	☺	☺	☺	☺	Investigates the effects of MPO inhibition on neuroinflammation
Varnas et al. 2019 [20]	☺	☺	☺	☺	☺	☺	☺	☹	☺	☺	☹	Correlates DAT and TSPO expression in patients with PD.
Jucaite et al. 2022 [21]	☺	☺	☺	☺	☺	☺	☺	☺	☺	☺	☺	May differentiate between MSA and PD.
Tarada et al. 2016 [22]	☺	☺	☺	☺	☺	☺	☺	☺	☺	☺	☺	It may explain patterns of neuroinflammation in early-stage disease.
Fang et al. 2022 [23]	☺	☺	☺	☺	☺	☺	☺	?	?	?	?	May provide an image-derived input function method for 18F-DPA-714.

☺ Low risk; ? Unknown; ☹ High risk.

## Data Availability

Not applicable.

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
