# Peer review of "PET Imaging of Neuro-Inflammation with Tracers Targeting the Translocator Protein (TSPO), a Systematic Review: From Bench to Bedside"

_diagnostics, 2023, doi:10.3390/diagnostics13061029_

Round 1

Author Response

First of all we would like to thank you for the attention shown and the precious suggestions which we have welcomed. According to your suggestions we have made the following corrections:

  • The title of the study has been changed
  • The naming of compounds has been corrected according to EANM’s guidelines
  • The asterisks in Figure 1 has been corrected
  • The word ”between-subject” in line 51 has been corrected into “inter-subject”
  • The word “clinical” in line 74 has been corrected into “Critical”
  • The superscripts indicating the isotopes have been corrected
  • The acronym BP in line 108 was defined
  • Vt has been corrected to Vt
  • The acronym MMSE in line 109 was defined

Reviewer 2 Report

The manuscript followed the PRISMA guideline with precise quality assessment, it provided the updated knowledge regarding the clinical PET imaging of TSPO. It’s a pity that the numerous typing errors and missing words affected the reading experience. The authors are suggested to recheck the words and spaces in every paragraph. There is no major question, but several points are addressed below: 

1. According to Instructions for Authors (Manuscript Preparation->Front Matter-> Title), the title should identify if the study reports (human or animal) trial data, or is a systematic review, meta-analysis or replication study. However, the title could not clearly identified that this manuscript is a systematic review, the authors are encouraged to mention the words "Systematic Review" in the title, and the "Type of the Paper" above the title is also encouraged to modified to "systematic review"

2. The Arabic numerals of radioisotopes should be written as superscripts, please recheck the typing. (18F in row 17 and 114, 11C in row 126, 152 etc.) 

3. The abbreviations should not appear without the full name at first mention. (MMSE in row 119, UPDRS in row 143, LEDD in row 147, MPO in row 175, MSA in row 184 etc.)

4. Please check row 147, did the authors mean "UPDRS" scores instead of the "UPRDS" scores ?

5. Except for the common radiopharmaceutical or TSPO targeting drugs, the authors are encouraged to simply drscript the targets or function of radiopharmaceutical at first mention. For example, the <superscripts>11C-CFT is reported to assess the severity of Parkinson's disease (PD) by binding to the striatal dopamine transporters (DATs)<reference>. Please add the brief descriptions of <superscripts>11C-PIB and <superscripts>11C-CFT. The description of <superscripts>18F-FE- PE21 is sufficient.

6. There are many unnecessary spaces in the paragraph, please recheck it.

7. There is no references cited in the discussion, however, the authors mentioned the other authors with their works in the discussion section indeed. Althought the references have been cited at other sections, the authors are encouraged to cite the references again in the discussion for better understanding to readers.

8. Although the manuscript focused on the TSPO PET imaging of clinical trials, I would suggest that the authors provide the information of drugs in developing. The authors are encouraged to talk about the current preclinical TSPO PET tracers in discussion such as <superscripts>11C-ER176,  <superscripts>18F-GE180, or other potential candidates.

Author Response

First of all we would like to thank you for your precious comments and considerations. According to your suggestions we have made the following corrections:

  • The title of the study has been changed
  • The superscripts indicating the isotopes have been corrected
  • The acronyms MMSE, UPDRS, LEDD, MPO, MSA were defined
  • The typing mistake UPRDS was corrected into UPDRS
  • [11C]PiB and [11C]-CFT were briefly described
  • The unnecessary spaces in the paragraphs were removed
  • References were cited in the discussion
  • 3rd generation radioligands including 11C-ER176 and 18F-GE180 were briefly described

Round 2

Reviewer 2 Report

The authors have addressed most of the problems and questions from the reviewers. In my opinion, the manuscript is improved to warrant publication in Diagnostics. However, the figure 1 (PRISMA flow-chart) is too blurry, please recheck the resolution of this figure. 

Author Response

We check the resolution of  figure 1. Thank you
